# Metatranscriptomic Analyses Reveal Important Roles of the Gut Microbiome in Primate Dietary Adaptation

**DOI:** 10.3390/genes14010228

**Published:** 2023-01-15

**Authors:** Mingyi Zhang, Xiaochen Wang, Ziming Wang, Shuxin Mao, Jiali Zhang, Ming Li, Huijuan Pan

**Affiliations:** 1School of Ecology and Nature Conservation, Beijing Forestry University, Beijing 100083, China; 2CAS Key Laboratory of Animal Ecology and Conservation Biology, Institute of Zoology, Beijing 100101, China; 3State Key Laboratory of Urban and Regional Ecology, Research Center for Eco-Environmental Sciences, Chinese Academy of Sciences, Beijing 100085, China; 4College of Life Sciences, University of Chinese Academy of Sciences, Beijing 100049, China

**Keywords:** diet, gut microbiome, primates, metatranscriptomic, frutivore, florivore, omnivore

## Abstract

The gut microbiome plays a vital role in host ecological adaptation, especially dietary adaptations. Primates have evolved a variety of dietary and gut physiological structures that are useful to explore the role of the gut microbiome in host dietary adaptations. Here, we characterize gut microbiome transcriptional activity in ten fecal samples from primates with three different diets and compare the results to their previously reported metagenomic profile. Bacteria related to cellulose degradation, like Bacteroidaceae and Alcaligenaceae, were enriched and actively expressed in the gut microbiome of folivorous primates, and functional analysis revealed that the glycan biosynthesis and metabolic pathways were significantly active. In omnivorous primates, Helicobacteraceae, which promote lipid metabolism, were significantly enriched in expression, and activity and xenobiotic biodegradation and metabolism as well as lipid metabolism pathways were significantly active. In frugivorous primates, the abundance and activity of Elusimicrobiaceae, Neisseriaceae, and Succinivibrionaceae, which are associated with digestion of pectin and fructose, were significantly elevated, and the functional pathways involved in the endocrine system were significantly enriched. In conclusion, the gut microbiome contributes to host dietary adaptation by helping hosts digest the inaccessible nutrients in their specific diets.

## 1. Introduction

Different dietary types are formed among species over a long evolutionary period, and they play a vital role in organisms to adapt to specific ecological niches [1]. Primates, as a taxon, closely related to humans and the most successful in evolution, are widely distributed, have highly diverse habitats and dietary characteristics [2], and provide a robust model system for studying dietary adaptions [3,4]. An increasing number of studies have shown that dietary adaptation is a key feature underlying primate evolution to different environments [5,6,7,8]. Thus, the study of primate dietary adaptation mechanisms can assist in elucidating how various species adapt to different environments.

Primates can be classified as folivorous (Fol, e.g., Colobidae), omnivorous (Omn, e.g., Cercopithecinae), or frugivorous (Fru, e.g., Hylobatidae) according to their nutritional requirements as each diet has a different major nutrient composition [9]. A folivorous diet contains a large amount of structural carbohydrates represented by fiber while a frugivorous diet is typically high in sugar [10,11,12]. Sichuan snub-nosed monkeys (*Rhinopithecus roxellana*), as typical folivorous colobus monkeys, primarily eat leaves and seeds as their major food source rather than fruits and insects, and they have a specialized foregut fermentation system that differs from other primates [13]. Their foregut fermentation system consists of a large and sacculated forestomach with a diverse assortment of microflora to digest the rough fibers [14]. The Rhesus macaque (*Macaca mulatta*) is the second-most widely distributed primate after humans, spanning both temperate and tropical latitudes and is representative of omnivorous primates [15]. Their dietary composition includes fruits, seeds, and flowers, as well as animal-based foods such as insects and bird eggs, reflecting their wide feeding range and high environmental adaptability [16]. Northern yellow-cheeked crested gibbon (*Nomascus annamensis*) are representative frugivorous primates that live in tropical and subtropical rainforests [17]. Fruits that have high concentrations of non-structural carbohydrates, mainly fructose, water, and lipids [12] are the main part of the gibbon diet (40–78%). Thus, in this study, these three primates were chosen as representatives of the various dietary adaptation mechanisms.

The effect of the microbiome on host health and development is gaining increasing attention, especially with the development of affordable and high-throughput DNA sequencing technologies [18]. Extensive research shows that the gut microbiome plays an important role in host dietary adaptation by affecting host metabolism and physiological status [19,20]. For instance, the gut microbiome can aid the host to flexibly and stably adapt to plant utilization by secreting enzymes (mainly cellulolytic and xylanolytic enzymes) [21] that are common in herbivores, particularly ruminants [22]; the survival of carrion eaters (i.e., scavengers) largely depends on their gut microbiomes’ ability to detoxify the toxic substances in carrion [23]; in insectivores, the utilization of protein can be improved by bacterial chitinases secreted by the gut microbiome [24]. The majority of these studies have focused on the spatial and temporal variation effects on the gut microbiota within the same host species. However, differences in the gut microbiome of different species with varied dietary adaptions within the same taxon group have rarely been examined. This might provide understanding of what drives host species to differentiate and occupy different ecological niches. Furthermore, most studies of the impact of different dietary composition on the gut microbiome have focused on short-term changes resulting from anthropogenic changes in diet [25]. Yet, host-microbe symbiosis is a tightly integrated whole that has coevolved over a long period of time [21], thus short-term changes in dietary composition may not accurately reveal this relationship. In addition, most of the current methods are limited to 16S RNA and metagenomic assays, which provide extensive information regarding the microbiota diversity and its potential function. However, there is no guarantee that the DNA detected originates from viable cells, whether the predicted genes are actually expressed, or under what conditions and to what extent the various gene expression occurs [26].

At present, the analysis of metatranscriptomics can offer novel insights into functional gene expression and microbial activities of the complex microbial communities at a specific time point or under specific environmental conditions [27]. However, how the gut microbiome affects the dietary adaption based on metatranscriptomics is still poorly understood [28]. Therefore, conducting metatranscriptomic analysis combined with metagenomic analysis can more clearly elucidate which genes are transcribed and identify the active metabolic pathways in bacterial communities that are associated with specific host or environmental features [26,29].

Here, we performed metatranscriptomic sequencing on primates with three types of diets (folivorous Sichuan snub-nosed monkeys [Fol, *n* = 3], omnivorous Rhesus macaque [Omn, *n* = 3], and frugivorous northern yellow-cheeked crested gibbon [Fru, *n* = 4]) and combined with our previous metagenomic (MG) data (Fol *n* = 10, Omn *n* = 10, and Fru *n* = 10) [11] to elucidate the mechanisms of dietary adaptation in primates with different dietary types. In this study, our aims were to elucidate the microbiome composition and metabolic strategies of different dietary types in primates and reveal how the microbiome interacts with the host, which can provide new insights into the mechanisms by which other animals and non-human primates adapt to their environment.

## 2. Materials and Methods

### 2.1. Sample Collection

Fecal samples were collected from primates with different dietary types including folivorous *Rhinopithecus roxellana* (*n* = 3, Fol) and omnivorous *Macaca mulatta* (*n* = 3, Omn) wild individuals in the Shennongjia National Park, Hubei Province. Fecal samples from frugivorous *Nomascus annamensis* (*n* = 4, Fru) were collected from captive individuals in Nanning Zoo, Guangxi Province. All fecal samples were collected in 15 mL centrifugal tubes immediately after animal defecation and added to RNAlater (Tiangen Biotech, Bejing, China), transported on dry ice to the laboratory of the Institute of Zoology, Chinese Academy of Sciences, and stored at −80 °C until processing.

### 2.2. RNA Extraction, Quality Control and Sequencing

Fecal RNA was extracted from RNAlater cryopreserved fecal samples using the Omega E.Z.N.A. Soil RNA isolation kit (Omega Bio-tek, Norcross, GA, USA), and DNase I (TaKara, Beijing, China) was used to remove genomic DNA. RNA quality was determined using a 2100 Bioanalyser (Agilent) and quantified using the ND-2000 (NanoDrop Technologies). High-quality RNA samples (OD 260/280 = 1.8–2.2, OD 260/230 ≥ 2.0, RIN ≥ 6.5, 28S:18S ≥ 1.0, >1 μg) were used to construct the sequencing library. The Ribo-Zero rRNA Removal Kit (Illumina, San Diego, CA, USA) was used for mRNA sequencing to selectively remove the rRNA and construct the transcriptome library. Sequencing was performed using the HiSeq 3000/4000 platform with at least 12 Gb per sample.

The raw paired-end reads were trimmed and quality controlled using SeqPrep (https://github.com/jstjohn/SeqPrep) to remove the adapter sequences at the 3′ and 5′ ends of the sequences and retain the reads greater than 50 bp in length after cutting. To retain high-quality pair-end reads, Sickle (https://github.com/najoshi/sickle) was used to remove reads that were less than 50 bp in length after clipping. The clean reads were separately aligned to the host genome (*R. roxellana*: assembly ASM756505v1; *M. mulatta*: assembly Mmul_10; *Nomascus leucogenys* assembly Asia_NLE_v1; and *Hylobates moloch*: assembly HMol_V2; the genome sequence for *N. annamensis* has not yet been published; *N. leucogenys* is in the same genus with *N. annamensis,* and these are closest between two species in phylogeny) using BWA [30]. To remove rRNA reads, the trimmed reads were compared with the rRNA databases SILVA SSU (16S/18S) and SILVA LSU (23S/28S) and filtered using SortMeRNA [31].

### 2.3. Construction of Non-Redundant Gene Sets

All clean data were assembled using Trinity bioinformatic software (version number: trinityrnaseq-r2013-02-25) [32] and transcripts longer than 300 bp were selected. Open reading frame (ORF) prediction was conducted using TransGeneScan software [33] for all transcript sequences obtained from the assembly. The predicted ORF sequences of all samples were clustered using CD-HIT (parameters: 95% identity, 90% coverage) [34]. The longest gene of each cluster was used as the representative sequence to construct the non-redundant gene set.

### 2.4. Taxonomic Annotation of RNA Non-Redundant Gene Sets

Non-redundant gene sets were used for BLASTP (BLAST Version 2.2.31+) searches and annotation against the NCBI Non-Redundant Protein Sequence (NR) database using an E-value cut-off of 10^−5^ [35] for taxonomic annotation of the microbiome.

The genes were taxonomically annotated using the corresponding taxonomic information obtained from the NR database. Then their expression abundance was calculated using the sum of gene expression abundances corresponding to the species. Gene expression abundance was calculated in reads per kilobase of transcript per million reads mapped (RPKM) using RSEM software [36]. Only taxonomic data with a domain classification of Bacteria or Archaea were kept for further analysis.

### 2.5. Functional Annotation of RNA Non-Redundant Gene Sets

Non-redundant gene sets were used for BLASTP searches and annotation against a Kyoto Encyclopedia of Genes and Genomes (KEGG) database using an E-value cut-off of 10^−5^ [35] for functional annotation of the microbiome. Functional analysis was performed using the KEGG orthology-based annotation system version 2.0 [37].

### 2.6. Statistical Analysis

The metagenomic taxonomic relative abundance (total 30 samples, Fol *n* = 10, Omn *n* = 10, and Fru *n* = 10) were obtained from previously published metagenomic sequence data from our lab [11].

The between-group differences were analyzed using the Kruskal–Wallis test and Dunn multiple comparisons test in the microbiome taxonomic using both metagenomic and metatranscriptomic data. Due to the greater number of differences in the metagenomic data, we chose *p* < 0.05 and *p* < 0.005 as thresholds for statistical significance for the metatranscriptomic and metagenomic data, respectively, to assess differences between microbiomes more accurately.

KEGG pathway differences among the different dietary groups in the metatranscriptomic data were analyzed using the Kruskal–Wallis test and Dunn multiple comparisons test; *p* < 0.05 was considered to be statistically significant.

Statistical analysis and charts (histograms) were performed and generated using GraphPad Prism 5 (GraphPad Software, CA, USA). Venn and line graphs were created using Hiplot (https://hiplot.org, accessed on 10 September,2022). Heatmaps were drawn using R package “pheatmap” (v.1.0.12).

## 3. Results

A total of 7,934,276,790 clean reads were obtained from the total RNA of all samples from the three dietary types of primates, and we obtained 661,555,446 non-rRNA reads after rRNA removal in the metatranscriptomic analysis (Figure 1A; Appendix A). After assembly, we obtained a total of 486,775 transcripts and 335,356 ORFs from all samples (Figure 1B; Appendix A). Based on taxonomic annotation of the results, we identified 8117 species from the fecal samples from all three dietary groups, which belonged to 122 phyla, 214 classes, 344 orders, 619 families, and 1965 genera (Appendix A). For functional annotation, we obtained 360 KEGG Level 3 pathways belonging to 46 KEGG Level 2 pathways (Appendix A). Based on our previous metagenomic data, we also identified 473 species belonging to 14 phyla, 24 classes, 38 orders, 59 families, and 110 genera from the taxonomic data (Appendix A). The composition of a metagenome represents the relative abundances of the taxa, whereas the composition of the metatranscriptome represents the relative activities of the taxa.

### 3.1. Different Taxa among Three Dietary Primates

The results revealed that 68 bacterial families from the metagenomic data and 402 bacterial families in the metatranscriptomic data among the gut microbiome differed among the primates of dietary groups (Kruskal–Wallis and Dunn tests MT *p* < 0.05 and MG *p* < 0.005) (Appendix A). Thirteen families showed overlap, including Acidaminococcaceae, Actinomycetaceae, Alcaligenaceae, Bacteroidaceae, Eggerthellaceae, Elusimicrobiaceae, Helicobacteraceae, Neisseriaceae, Porphyromonadaceae, Puniceicoccaceae, Rhodospirillaceae, Streptococcaceae, and Succinivibrionaceae (Figure 2A).

Based on analyses of metagenomics and metatranscriptomics, the distribution heatmap of the thirteen families indicated the relative abundance (Figure 2B) and the relative activity (Figure 2C) of the bacterial taxa, respectively. Figure 3 illustrates that most of the thirteen bacterial families in the metagenomic and metatranscriptomic analyses demonstrated consistent trends in relative abundance and relative activities.

The distribution of the thirteen families in the gut microbiome of three dietary groups of primates is presented in the heatmaps (Figure 2B,C). Figure 2B depicts the heatmap showing the relative abundance of bacterial taxa according to the metagenomic analysis, while Figure 2C depicts the heatmap showing the relative activity of the bacterial taxa according to the metatranscriptomic analysis. The line graphs depicting the abundance in the thirteen families in both the metagenomic and metatranscriptomic analyses showed that most of them demonstrated consistent trends in both relative abundance and relative activities (Figure 3).

We found that Bacteroidaceae and Alcaligenaceae were significantly enriched in folivorous primates in both the metagenomic and metatranscriptomic analyses (Figure 3A,D). Bacteroidaceae directly promotes cellulose digestion [38], while Alcaligenaceae promotes cellulose digestion indirectly by promoting the growth of Bacteroidaceae [39]. A similar trend occurred in Puniceicoccaceae and Rhodospirillaceae, as both exhibited lower relative transcriptional activity than relative abundance (Figure 3F,H) in folivorous primates, indicating that certain factors at the time of sampling or brief dietary changes may have led to transient lower activity [40]. In addition, we found that there was a significantly reduced number of the dominant bacterial family in omnivorous primates than in the other dietary groups. Only Helicobacteraceae were enriched in both the metagenomic and metatranscriptomic analyses (Figure 3I). We suggest that this is because they have a more diverse dietary consumption, and, therefore, omnivorous primates have fewer dominant bacterial species than the specialized primates.

In frugivorous primates, Elusimicrobiaceae, Neisseriaceae, and Succinivibrionaceae were enriched in both the metagenomic and metatranscriptomic analyses (Figure 3B,C,E); these families are also significantly enriched in hosts consuming high-sugar diets [41,42,43]. In addition, Acidaminococcaceae showed similar dominance over other species, but its relative activity was less than its relative abundance (Figure 3J). In addition, we found trends in both the metagenomics data as well as the metatranscriptomic data for Actinomycetaceae, Eggerthellaceae, and Porphyromonadaceae that were inconsistent. Of these, the relative abundance of Actinomycetaceae and Eggerthellaceae were enriched in frugivorous primates, but their relative activities were enriched in folivorous primates (Figure 3L,M). The relative abundance of Porphyromonadaceae was enriched in frugivorous primates, but its relative activity was enriched in omnivorous primates (Figure 3K).

These results indicate that the respective characteristic gut microbiome of the three dietary groups of primates showed significant differences in both the metagenomics and metatranscriptomics analyses, but the trend patterns in two analytical methods for the same diet type were mostly consistent. Accordingly, we speculated that these differences in microbiota and their activity may make a unique contribution to the dietary adaptive capacity of the host.

### 3.2. Differences in Gut Microbiome Function among Primates with Three Different Diets

The results of the pathway categorization using the KEGG pathway database are shown in Figure 4. These results show that the gut microbiome was functionally enriched for metabolic pathways (KEGG Level 1) and that the relative abundance of metabolic pathways contributed to half of the total functional pathway abundance (Fol 50.09%, Omn 43.81%, Fru 41.05%). The metabolic pathways that were specifically identified were the carbohydrate, energy, amino acid, and cofactors and vitamins metabolism pathways (KEGG Level 2) (Figure 4).

In addition, the analysis of differences among these KEGG pathways showed that there were ten KEGG level 2 pathways with significant differences among the three dietary groups of primates (Dunn Test *p* < 0.05) (Figure 5). We found that these pathways were related to the sensory system and glycan biosynthesis and metabolism that was significantly enriched in folivorous primates (Figure 5A,C), and that differences related to glycan biosynthesis and metabolism were especially prominent. Glycans are mainly provided by dietary plants and are the main nutrients for folivorous animals; glycans cannot be hydrolyzed by the digestive enzymes of animals and thus are the main substrates of bacterial fermentation in the gut [44,45] (Figure 5C). Pathways related to the circulatory system, drug resistance and antineoplastic activity, substance dependence, xenobiotic biodegradation and metabolism, and lipid metabolism were significantly enriched in omnivorous primates (Figure 5D–H). Pathways involved in the endocrine system, membrane transport and nervous system were significantly enriched in frugivorous primates (Figure 5B,I,J).

## 4. Discussion

Many studies have shown that the composition and function of the gut microbiome are closely related to the evolutionary history and dietary strategy of the host [46,47]. Host diets and their evolutionary history are often correlated (i.e., closely related species might share similar diets); it is challenging to disentangle the effect of host diet composition on the gut microbiome from other influencing factors [48,49]. This might lead to controversy regarding the contribution of gut microbiome to host dietary adaptations. In this study, we performed metatranscriptomic sequencing of the gut microbiome of three primates with different dietary types and combined the results with previously reported metagenomic data for dual differential validation. The results showed that different dietary primates have significant difference both in microbiome presence and expression. Most of these bacteria, which differ significantly in different dietary primates, are consistent in abundance and expression activity in the same species. Thus, we suggest that the gut microbiome of primates with different dietary types differentially assist their hosts to utilize the ingested fibers or plant secondary compounds, protein, and fructose or to cope with xenobiotic biodegradation in complex environments. Our results support the proposal that the gut microbiome plays a key role in host dietary adaptation. Such findings illuminate how the gut microbiome facilitates dietary adaptation in primates with different diets and assists them to adapt to their best dietary ecological niches.

The gut microbiome of folivorous primates reflects their unique diet type. We observed numerous bacteria with metabolic pathway activations associated with complex structural carbohydrates (such as cellulose and lignin). These bacteria included Bacteroidaceae, Alcaligenaceae, Puniceicoccaceae, and Rhodospirillaceae (Figure 3), and functional pathways included glycan biosynthesis and metabolism (Figure 5C). Bacteroidaceae can contribute to the degradation of cellulose both in African large-herbivorous wildlife [50] and experimental mice [51]. Alcaligenaceae can promote the growth of Bacteroidaceae by impacting estrogen metabolism [52], thus indirectly contributing to the feeding adaptation of folivorous primates [39]. Likewise, Rhodospirillaceae are taxa that are enriched in the microbiome of animals with high-fiber diets [53]. Compared with omnivorous and frugivorous primates, the metatranscriptomics results indicated that folivorous primates were enriched for the activation of functional pathways involved with glycan biosynthesis and metabolism (Figure 5C). These pathways were also enriched in the gut microbiomes of wild giant pandas (*Ailuropoda melanoleuca*) [54] and Tibetan macaques (*Macaca thibetana*) [55]. These results suggest that the unique gut microbiome in folivorous primates may allow them to efficiently digest cellulose and hemicellulose to produce glycans, which is beneficial to the host by obtaining energy from low quality food (high fiber, low energy). Puniceicoccaceae was significantly enriched in the gut microbiomes of folivorous primates (Figure 3H); its functional role in folivorous primates should be further studied although they are typically associated with nitrogen-fixation [56].

Macaques, as representative omnivorous primates, alternate between a large number of nutrients (proteins, fats, and carbohydrates) as energy sources as a strategy to tolerate interannual variation [57], compared to other dietary primates. The results showed that Helicobacteraceae were significantly enriched in the gut microbiome of macaques (Figure 3I) as well as in mice fed a high-fat diet [58]. Meanwhile, we found that high activity of lipid metabolic pathways in their gut microbiome confirmed the adaptation to their high-protein and high-fat diet (Figure 5H). In our previous study, we found a high number of bacteria involved in lipid metabolism in the gut microbiome of omnivorous macaques [11]. This may explain how macaques efficiently digest and absorb these fat and protein rich diets through their gut microbiome. Other literature has reported that continued habitat destruction could lead to a marked reduction in the hosts’ detoxification capacity and adaptive capacity in red colobus monkeys (*Procolobus gordonorum*) [59]. The results demonstrate that xenobiotic biodegradation and metabolism activity are significantly enriched in the gut microbiome of omnivorous primates (Figure 5G). Xenobiotics are defined as chemicals to which an organism is exposed but are extrinsic to the normal metabolism of that organism [60]. The gut microbiome can change the chemical structure of xenobiotics, thereby altering its effective duration, biological effectiveness, and biological effects [61], which indicates that omnivorous primates can utilize some foods that contain toxic xenobiotics. We speculate that the possible reason for the enrichment of the xenobiotic biodegradation and metabolism pathways in omnivorous primates is to increase the ability to detoxify toxic substances present in their varied diet. Thus, further xenobiotics-related studies relating to dietary composition may increase our knowledge of the effects of habitat contamination on the adaptive changes in wildlife.

As fruit-eating specialists, there are high concentrations of total non-structural carbohydrates in the diets of frugivorous primates, especially fructose but also pectin and water [12]. We hypothesize that their gut microbiome would be aligned with the high amount of pectin and fructose in their diet, and the results confirmed this hypothesis. We found significant abundance and expression activity of the Neisseriaceae and Succinivibrionaceae families in frugivorous primates (Figure 3C,E). Succinivibrionaceae play an essential role in fruit-eating animals [43] as studies show that pectin can stimulate an increase in Succinivibrionaceae [62] in the gut. Neisseriaceae is also more abundant in frugivorous than insectivorous bats [63]. Meanwhile, our functional exploration revealed that pathways of the endocrine system were significantly enriched in frugivorous primates (Figure 5I). It has been shown that microbiome metabolites act as endocrine factors that can influence host glucose tolerance and insulin sensitivity, ultimately affecting the host’s glucose metabolism and energy balance [64]. We suggest that this contributes to the adaptation of fruit-eating primates to a high-sugar, high-energy structure diet. Although the transcriptional expression of Acidaminococcaceae was slightly lower in this study, we found that Acidaminococcaceae were enriched in frugivorous primates (Figure 3J), indicating that Acidaminococcaceae could contribute to the adaption to a high-sugar diet [65]. In addition, the bacterial transcriptional activity also demonstrated that the activity of the host’s gut microbiome, especially in the dominant group, was dynamic as indicated by metatranscriptomic analyses [66]. These data demonstrate a flexible contribution of the gut microbiome to the host during dietary adaptation.

Compared with previous studies, this is the first to report that the gut microbiome of primates with different diets differs significantly, and our results were verified by dual differential validation through metagenomic and metatranscriptomic techniques. Most of the differences in bacterial transcriptional activity were consistent with the differences in bacterial abundance. In contrast to previous studies that focused only on the contribution made by the dominant microbiota groups in dietary adaptation, we found that microbiota with a significantly lower abundance (most differential microbiota families had a relative abundance of <1%) (Appendix A) were associated with functional activity indicating their contribution to host dietary adaptation.

## 5. Conclusions

Host–symbiont co-evolution is essential for adaptation to the environment, especially regarding dietary adaptation [21]. In folivorous primates, the gut microbiome was enriched for bacteria as well as activity of metabolic pathways associated with complex structural carbohydrates such as cellulose and lignin. Conversely, the gut microbiome in frugivorous primates was associated with the digestion of non-structural carbohydrates, especially fructose as well as pectin. Although the gut microbiome of omnivorous primates appeared to have fewer characteristic bacteria, transcriptional gene expression activity analysis indicated that the gut microbiome of these primates had higher protein and lipid degradation capacity. Comparisons among these microbiomes revealed that the differences observed reflect the host’s dietary ecological niche, suggesting a significant role of the host-gut microbiome association in host dietary adaptation. Therefore, we suggest that the elucidation of dietary adaptation mechanisms of species is important not only for the conservation of targeted species, but also for the protection of their habitat.

## Figures and Tables

**Figure 1 genes-14-00228-f001:**
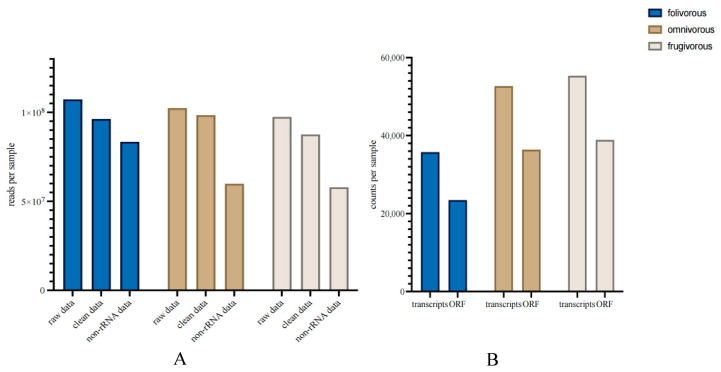
Effect of data decontamination and assembly of the gut microbiome metatranscriptomic analyses. (**A**) Read count for microbiome metatranscriptome before and after data decontamination, per primate dietary type fecal sample. (**B**) Transcript count per primate dietary type fecal sample for the assembled results and predicted open reading frames.

**Figure 2 genes-14-00228-f002:**
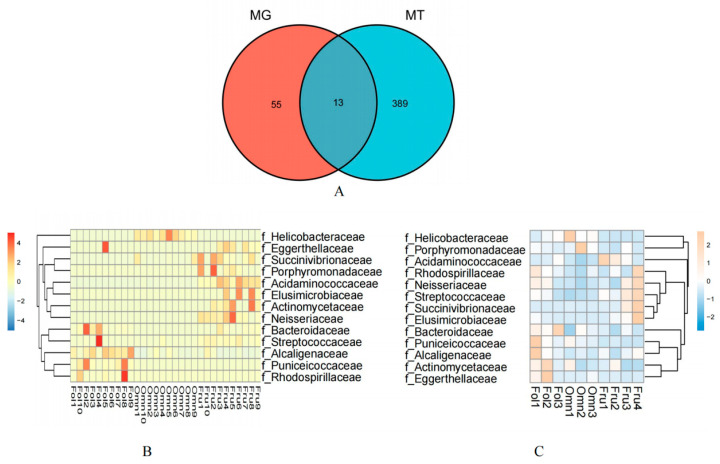
Association of taxonomic families in the gut microbiome of primates with differing diets. (**A**) The Venn diagram shows the numbers of significantly different bacteria families that were shared or unshared by metagenomic and metatranscriptomic analysis. Heatmap of Spearman’s rank correlation coefficients between primates with different dietary consumption and associated microbial species identified in fecal gut metagenomic (**B**) and metatranscriptomic (**C**) analyses. Red and blue colors denote positive and negative correlations, respectively. Color intensity is proportional to Spearman’s rank correlation values.

**Figure 3 genes-14-00228-f003:**
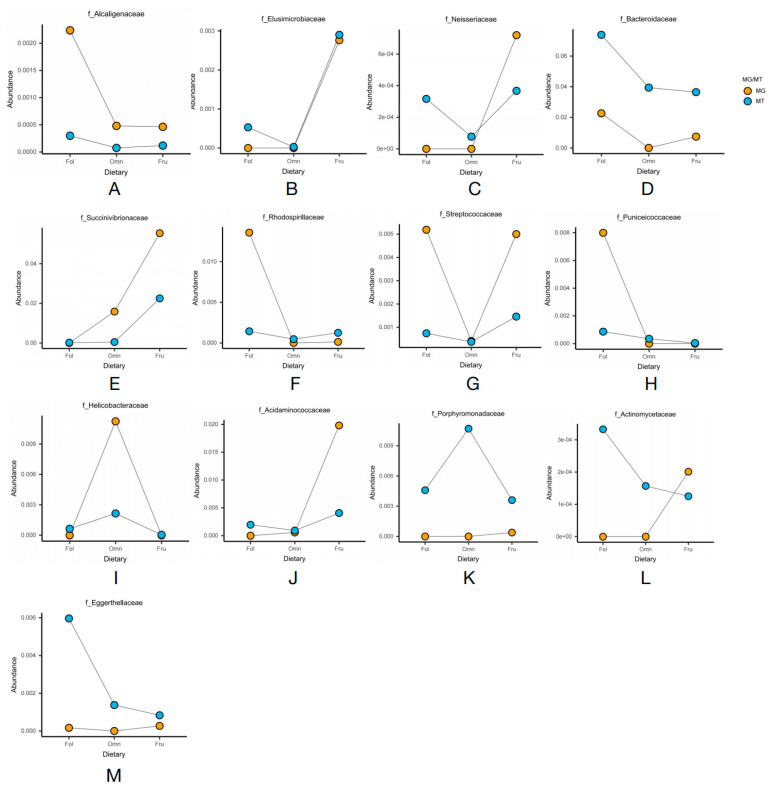
Line graphs of the relative abundance of microbiome family levels in the metagenomic and metatranscriptomic analyses. (**A**) Alcaligenaceae; (**B**) Elusimicrobiaceae; (**C**) Neisseriaceae; (**D**) Succinivibrionaceae; (**E**) Bacteroidaceae; (**F**) Rhodospirillaceae; (**G**) Streptococcaceae; (**H**) Puniceicoccaceae; (**I**) Helicobacteraceae; (**J**) Acidaminococcaceae; (**K**) Porphyromonadaceae; (**L**) Actinomycetaceae; (**M**) Eggerthellaceae.

**Figure 4 genes-14-00228-f004:**
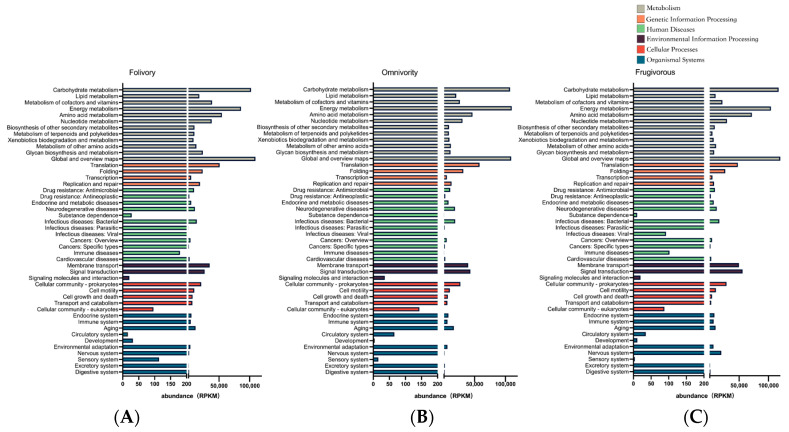
Kyoto Encyclopedia of Genes and Genomes (KEGG) pathway classifications abundance in reads per kilobase of transcript per million reads mapped (RPKM) for the gut metatranscriptome in folivores (**A**), omnivores (**B**), and frugivores (**C**).

**Figure 5 genes-14-00228-f005:**
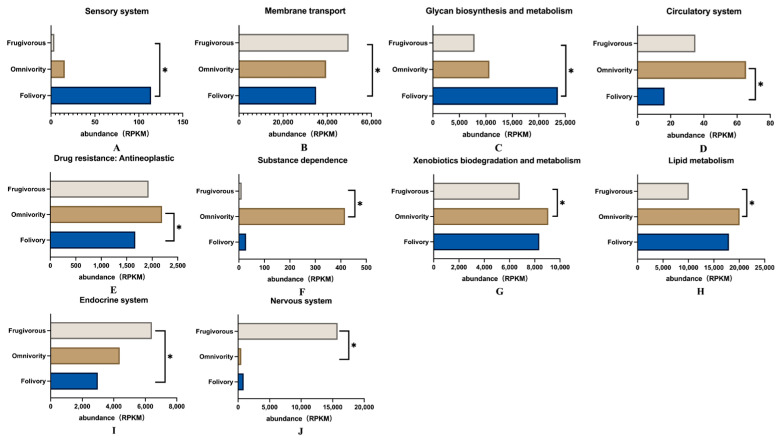
Differential functional gene expression abundance in RPKM of KEGG Level 2 pathways in the gut microbiome of primates with three different diets. (**A**) Sensory system; (**B**) Membrane transport; (**C**) Glycan biosynthesis and metabolism; (**D**) Circulatory system; (**E**) Drug resistance: Antineoplastic; (**F**) Substance dependence; (**G**) Xenobiotics biodegradation and metabolism; (**H**) Lipid metabolism; (**I**) Endocrine system; (**J**) Nervous system. Using the Kruskal–Wallis and Dunn tests, * *p* < 0.05 was considered to be statistically significant.

## Data Availability

Sequencing data can be found in the Sequence Read Archive (ncbi.nlm.nih.gov/sra, accessed on 14 January 2020) under BioProject PRJNA923763.

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
