# Peer review of "Metatranscriptomic Analyses Reveal Important Roles of the Gut Microbiome in Primate Dietary Adaptation"

_genes, 2023, doi:10.3390/genes14010228_

Round 1
Reviewer 1 Report
In this paper, the authors studied transcriptional activity in ten fecal samples from primates with three different diets and compare the results to their previously reported metagenomic profile. This paper is built on the authors’ previous work; its impact might be limited. But overall, the manuscript is well-written. The comments below should be addressed.
Comments:
1),
I have some complaints is the English writing. There are some grammar mistakes or incurate usage. I listed several of them below, but not enumerating all of them. Careful proofreading is needed.
[Line 66]
[line 218] The distribution heatmap of the thirteen families indicating the relative abundance of the bacterial taxa according to the metagenomic analysis (Figure 2B) and the relative
The authors sometimes use long, cumbersome sentences. It impairs the readability of the manuscript.
2),
Figure 2 legend can be shortened. The Venn diagram is straightforward; the texts seems redundant.
The authors should add color scale bar regarding the intensity.
3),
[line 134] “the genome sequence for N. annamensis has not yet been published, We selected the genome of its close relatives) using BWA [30]”
Some additional technical details might be needed. How close are the two species? Will it impact the transcriptomic analysis results?
Author Response
Respond to Review1
In this paper, the authors studied transcriptional activity in ten fecal samples from primates with three different diets and compare the results to their previously reported metagenomic profile. This paper is built on the authors’ previous work; its impact might be limited. But overall, the manuscript is well-written. The comments below should be addressed.
Comments:
- I have some complaints is the English writing. There are some grammar mistakes or incurate usage. I listed several of them below, but not enumerating all of them. Careful proofreading is needed.
[Line 66][line 218] The distribution heatmap of the thirteen families indicating the relative abundance of the bacterial taxa according to the metagenomic analysis (Figure 2B) and the relative
The authors sometimes use long, cumbersome sentences. It impairs the readability of the manuscript.
Response: Sorry for the language problem, we had improved the language using a Enago (https://www.enago.com) for language editing.
- Figure 2 legend can be shortened. The Venn diagram is straightforward; the texts seems redundant. The authors should add color scale bar regarding the intensity.
Response: We appreciate the reviewer’s comments. We have reduced the legend of Figure 2. We are sorry that we did not showing that the color scale bar in our manuscript. This is an oversight on our part. The corresponding changes have been adjusted, we have added color scale bar regarding the intensity in Figure 2(B, C).
- [line 134] “the genome sequence for annamensis has not yet been published, We selected the genome of its close relatives) using BWA [30]”
Some additional technical details might be needed. How close are the two species? Will it impact the transcriptomic analysis results?
Response: We appreciate the reviewer’s comments. N. annamensis belongs to the same genus as N. leucogenys and is a new species described by Christian Roos (Vietnamese Journal of Primatology, 2010) in 2010. We selected the host genome of N. leucogenys for removal, which is the closest related host genome we could find to N. annamensis. The corresponding changes were made in line173-174. We believe that this does not affect the results of the transcriptome analysis, we used BLASTP (BLAST Version 2.2.31+) searches and annotation against the NCBI Non-Redundant Protein Sequence (NR) database using an E-value cut-off of 10-5, this enables very accurate annotation of microbiome sequences, and because microbiome and primate sequences are so different, this approach is virtually unaffected by the remaining host genome.
Reviewer 2 Report
Excellent and well written manuscript. However some findings were not tally with the analysis and conclusion. Use Spearman's rank correlation coefficient to support your hypothesis, on the microbiome's role in co-evolution with the host, particularly dietary adaptations. I don't see any correlation analyses in your results to support your conclusion above.

Author Response
Respond to Review 2
Comments and Suggestions for Authors
Excellent and well written manuscript. However, some findings were not tally with the analysis and conclusion. Use Spearman's rank correlation coefficient to support your hypothesis, on the microbiome's role in co-evolution with the host, particularly dietary adaptations. I don't see any correlation analyses in your results to support your conclusion above.
- [line96-101] Why are sample sizes different for metatranscriptomic and metagenomic analyses of primates' three dietary types?
Response: We appreciate the reviewer’s comments. Compared to metagenomic, metatranscriptomic require higher quality samples, and a large proportion of the samples we collected were not successful in construct the transcriptome library, and our sample size should meet these analyses according to the minimum statistical requirements. In addition, we combined with metagenomic data for exploring the role of gut microbiome in host dietary adaptation.
- [line110-112] Why are only 3 to 4 samples of individuals being studied? Is it following animal ethics protocols, or are only three individuals available for research study?
Response: We appreciate the reviewer’s comments. We are in compliance with animal ethics protocols, and as mentioned above, we collected many fecal samples, and due to the high quality of samples required for the metatranscriptomic, some of them did not construct the transcriptome library successful, so we abandoned them. In addition, as the Sichuan golden snub-nosed monkeys and macaques were wild individuals, it was not easy to track them in the wild and collect feces. While gibbons were critically endangered in the wild and captive individuals in zoos were rare. Given the experimental period and other factors, we finally obtained only ten samples.
- [line224-231] How can findings be compared when both analyses have different data sets? Briefly explain whether or not normalisation was performed for the number of primates used in both analyses.
Response: We appreciate the reviewer’s comments. Since metagenomic and metatranscriptomic are different types of data, we only compare the taxonomic relative abundance of the metatranscriptomic with previously published metagenomic data. We plotted line graphs to better observe the trends and explore the relationship between microbiome abundance and microbiome expression activity. Considering the large differences between the sequence data types and sample sizes, we did not combine analyses them, since we mainly explored trends in abundance and then we did not normalize primate numbers.
- [line255-259]This is only a suggestion because no correlation analysis between bacterial families and their abundances was performed, correct?
Response: We appreciate the reviewer’s comments. Yes, this is only a suggestion, we have discussed the results obtained in the two data sets separately. We did not analyze the correlation between their bacterial abundance and bacterial activity, because we thought that many factors could result in this phenomenon. This is an interesting finding that we will probably continue to research further next, so we removed the corresponding expressions in line279.
- [line317-318] Use Spearman's rank correlation coefficient to support your hypothesis, on the microbiome's role in co-evolution with the host, particularly dietary adaptations. I don't see any correlation analyses in your results to support your conclusion.
Response: We appreciate the reviewer’s comments. We are sorry that the expression is not accurate. Our intention was to express that the gut microbiome played an important role in the host's dietary adaptation, and we did not use correlation coefficient to support our hypothesis. In addition, the “co-evolution” was not used correctly and the expression may be inaccurate here, so we removed the corresponding expressions in [line338-lin339].